# Birds land reliably on complex surfaces by adapting their foot-surface interactions upon contact

William RT Roderick[†]*, Diana D Chin[†], Mark R Cutkosky, David Lentink*

Department of Mechanical Engineering, Stanford University, Stanford, United States

**Abstract** Birds land on a wide range of complex surfaces, yet it is unclear how they grasp a perch reliably. Here, we show how Pacific parrotlets exhibit stereotyped leg and wing dynamics regardless of perch diameter and texture, but foot, toe, and claw kinematics become surface-specific upon touchdown. A new dynamic grasping model, which integrates our detailed measurements, reveals how birds stabilize their grasp. They combine predictable toe pad friction with probabilistic friction from their claws, which they drag to find surface asperities—dragging further when they can squeeze less. Remarkably, parrotlet claws can undergo superfast movements, within 1–2 ms, on moderately slippery surfaces to find more secure asperities when necessary. With this strategy, they first ramp up safety margins by squeezing before relaxing their grasp. The model further shows it is advantageous to be small for stable perching when high friction relative to normal force is required because claws can find more usable surface, but this trend reverses when required friction shrinks. This explains how many animals and robots may grasp complex surfaces reliably.

DOI: https://doi.org/10.7554/eLife.46415.001

*For correspondence:
wrtr@stanford.edu (WRTR);
dlentink@stanford.edu (DL)

[†]These authors contributed equally to this work

## Introduction

While airplanes require runways and most aerial robots need flat landing areas, arboreal birds consistently land on perches with a wide range of geometries and surface properties. Recent advances in perching capabilities of aerial robots have improved their landing ability on engineered surfaces. However, they still lack the versatility and robustness that birds exhibit when landing on an extraordinarily wide range of complex engineered and natural surfaces (*Kovac, 2016*; *Roderick et al., 2017a*; *Doyle et al., 2013*). Previous studies have quantified leg and wing dynamics (*Provini et al., 2012*; *Provini et al., 2014*; *Chin and Lentink, 2017*; *Provini and Abourachid, 2018*) and suggested visual control strategies (*Chin and Lentink, 2017*; *Lee et al., 1993*) that arboreal birds use during landings. However, it is unclear whether the same findings would apply with perches that have different geometric properties and challenging surface textures.

Even less is understood at the actual interface between the bird's foot and the perch surface. While previous studies have examined the morphology of bird feet in cadavers (*Hopson, 2001*; *Sustaita et al., 2013*) and claw shape in relation to body size and lifestyle (*Pike and Maitland, 1999*; *Birn-Jeffery et al., 2012*; *Feduccia, 1993*), no in vivo studies quantify toe forces during grasping or how claws actually interact with perch surface features. Consequently, we do not know how birds actually grasp perches so reliably in a seemingly effortless manner.

To elucidate the mechanisms and strategies responsible for the versatile perching capabilities of arboreal birds, we studied voluntary landings made by Pacific parrotlets (*Forpus coelestis*) on perches with different geometries and textures (*Video 1*). We tested nine landing perch variations in total, which included three natural variations, three diameter variations, and three man-made surface variations (*Figure 1a*). The natural perches were constructed from ~0.75" (19 mm) diameter

**eLife digest** Most of the flying vehicles designed by humans need to land on smooth, standardized surfaces such as runways. A bird, on the other hand, can use structures that vary widely in diameter and texture, from phone lines to branches to statues. Yet, few studies have focused on how these animals transition from the air to a perch, and especially on how they adapt to different surfaces.

To fill this gap, Roderick, Chin et al. recorded how Pacific parrotlets landed on nine natural and man-made perches that varied in diameter and texture, ranging from smooth Teflon to rough sandpaper. High-speed cameras tracked each of the landings while sensors measured how hard the birds landed on and squeezed the perches.

The experiments revealed that the first landing phase was the same regardless of the nature of the perch. The birds used their wings to slow down, unfurled their feet and claws in preparation for touchdown and then allowed their legs to absorb the landing impact.

Once the feet had made contact with the surface, however, the birds used their toes and claws to adapt to different perches. First, they steadied their grasp by tightly squeezing the perches. Then, the parrotlets dragged their claws on the surface of the perches to find minuscule bumps and dips that allowed better stabilization. These movements could be remarkably fast – in the range of one to two milliseconds. The birds also curled their claws more on perches that were harder to grasp. Once secured on the branch, they relaxed their grip.

The results by Roderick, Chin et al. will help biologists understand how birds, insects and even large tree-dwelling creatures can grab perches in various environments. This knowledge will also be relevant for engineers who are trying to create robots that can climb or land on diverse surfaces.

DOI: https://doi.org/10.7554/eLife.46415.002

branches from three different trees. The first tree was a Coast Live Oak (*Quercus agrifolia*), which varies in roughness at different parts of its branches. The second was a Floss-Silk tree (*Ceiba speciosa*), which has relatively smooth, soft branches and is native to the same forests of Ecuador and Peru (*Nasr et al., 2018*) as the Pacific parrotlets. The third was a Sweet Olive tree (*Osmanthus fragrans*), which has rough, striated branches. Birch dowels, which are commonly used for commercial bird housing perches (our parrotlet housing uses 0.625"-dia. birch dowels), were used to construct the three diameter variations: 1.5" (38 mm), too large for the parrotlets to wrap their feet more than a quarter of the way around; 0.75" (19 mm), which their feet can wrap about halfway around; and 0.25" (6 mm), which their front and rear toes wrap around completely and overlap. To test surface texture extremes, three 0.75" diameter perches were constructed by wrapping birch wooden dowels in foam (squishy), Teflon (slippery), and sandpaper (rough).

To characterize the surface properties of these perches, we made 3D structured-light scans to quantify roughness (*Supplementary file 1A*) and reconstruct surface profiles (see Materials and methods). The profiles show the variation in surface roughness, from smoothest, Teflon, to roughest, sandpaper (*Figure 1A*). To evaluate the effect of these properties on foot-surface interactions, we conducted toe and claw drag tests to measure friction forces on each surface. We also performed claw indentation tests to measure surface deformation (see Materials and methods; *Figure 1B*). Finally, we made high-speed recordings of parrotlets landing on the nine different perches. The perches were split in half and instrumented on a pair of force/torque sensors for measuring net and squeeze forces exerted by their feet (see Materials and methods; *Figure 1C*).

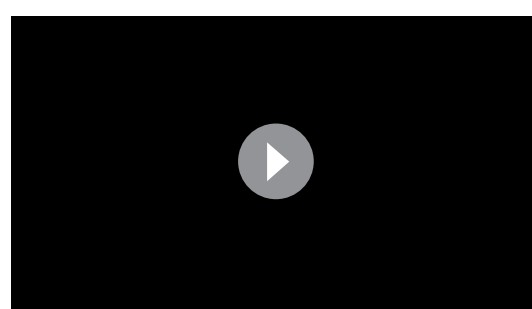

**Video 1.** Representative landings on the nine tested perches.
DOI: https://doi.org/10.7554/eLife.46415.004

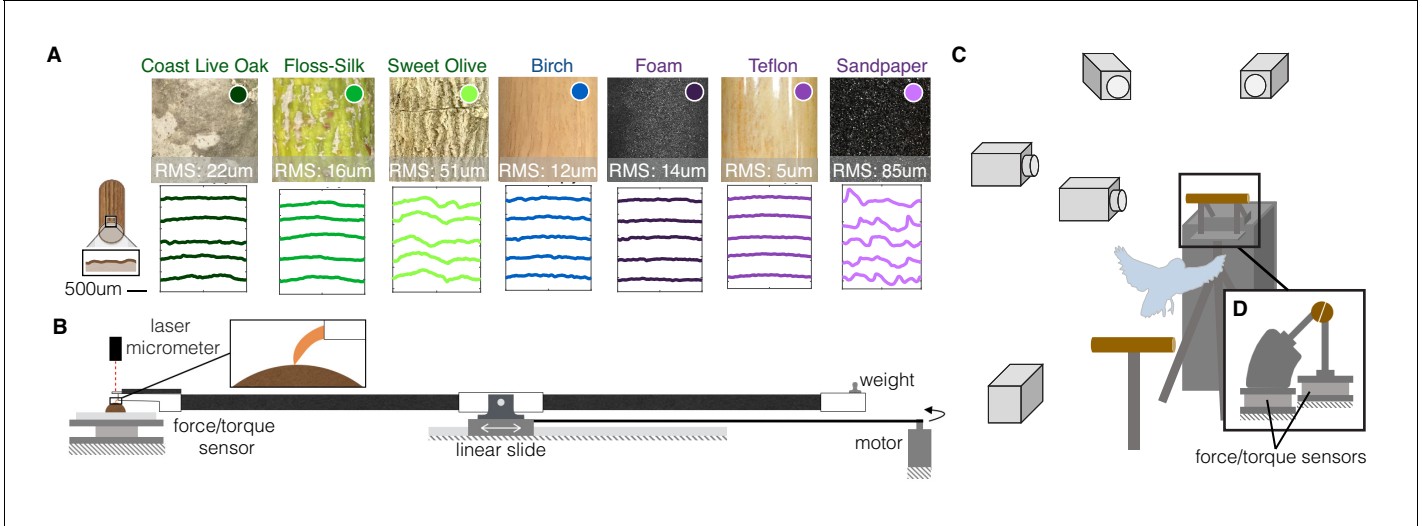

**Figure 1.** Experimental setups to measure the dynamics, kinematics, and surface interactions that birds use to land on a wide range of surfaces. (**A**) Perches of 7 different natural or human-made surfaces were tested. RMS roughness values and circumferential surface profiles (five representative profile segments are shown for each perch) were derived from 3D scans of each surface. (**B**) To quantify foot-surface interactions, sections of each perch surface were affixed to a force sensor for measuring friction forces during claw and toe drag tests. A laser micrometer was used to measure the depth of surface deformations during claw indentation tests. (**C**) To understand how birds adapt to these variable substrates, high-speed recordings were made during voluntary landings of Pacific parrotlets (*Forpus coelestis*) on a custom-built split perch (**D**). The front and back halves of each perch were mechanically isolated and instrumented by separate force/torque sensors. The split was oriented at 20 degrees from the vertical to best align with the center of the parrotlets' feet and therefore isolate front and rear toe forces. This design enabled us to recover squeeze forces exerted by the feet in addition to the total landing forces (see Materials and methods). (**B**) and (**D**) are drawn proportionally to scale.
DOI: https://doi.org/10.7554/eLife.46415.003

## Results

### Stereotyped wing and leg dynamics

The parrotlets exhibit a stereotyped set of landing behaviors across all perch variations (*Figure 2A*, *Video 2*). Landing is initiated by braking with the wings ('aerial'), and then the legs absorb the remaining momentum upon impact with the perch ('absorption'). Both feet would make contact with the perch within a few milliseconds of each other, but initial contact tended to be led by one preferred foot (bird 1 = 100% right foot, bird 2 = 83% left, bird 3 = 85% left). The parrotlets then wrap their toes and claws more securely around the perch to establish a static grasp ('anchoring'). After this stage, they often take additional steps (2.0 ± 0.7, n = 78 landings) to adjust their footing ('adjustments'). The birds would occasionally overshoot or undershoot the perch, leading to more variance in the foot angle at which they establish a static grasp (*Figure 2B*). However, while the net leg force magnitudes (*Figure 2B*) and directions (*Figure 2C*) also exhibit some variance, the average landing force trends remain remarkably similar across the different perches.

The variation seen during landing may be explained by the parrotlets' landing strategy. According to Tau theory (*Lee et al., 1993*), birds control their landings by visually estimating their time to contact, $\tau(t)$. To land, they adjust their approach speed to maintain a constant $\dot{\tau}(t)$, the time rate of change of tau. More specifically, tau is defined as the distance to the perch $s$ divided by the speed of approach $v$, and is therefore a first-order approximation for the time to contact. If a bird brakes with constant deceleration $a$, then $\tau = \frac{s}{v} = \frac{0.5at^2}{at} = 0.5t$, and $\dot{\tau}(t) = 0.5$. If $\dot{\tau}(t) < 0.5$, braking is decreasing until the bird stops at the landing perch, and if $0.5 < \dot{\tau} < 1$, braking is increasing until the bird makes a controlled collision with the landing perch (*Lee et al., 1993*). Based on their aerial kinematics, the parrotlets maintain relatively constant $\dot{\tau}$ values (*Figure 2E*) consistent with those of controlled collisions (*Figure 2F*). The smallest landing target, the 0.25"-dia. perch, had the smallest average $\dot{\tau}$ value of 0.80. The foam perch had the highest at $\dot{\tau} = 0.94$, which indicates that parrotlets increase their braking more when they are closer to the softer perch. Across all perches, the high $\dot{\tau}$ values (>0.5)

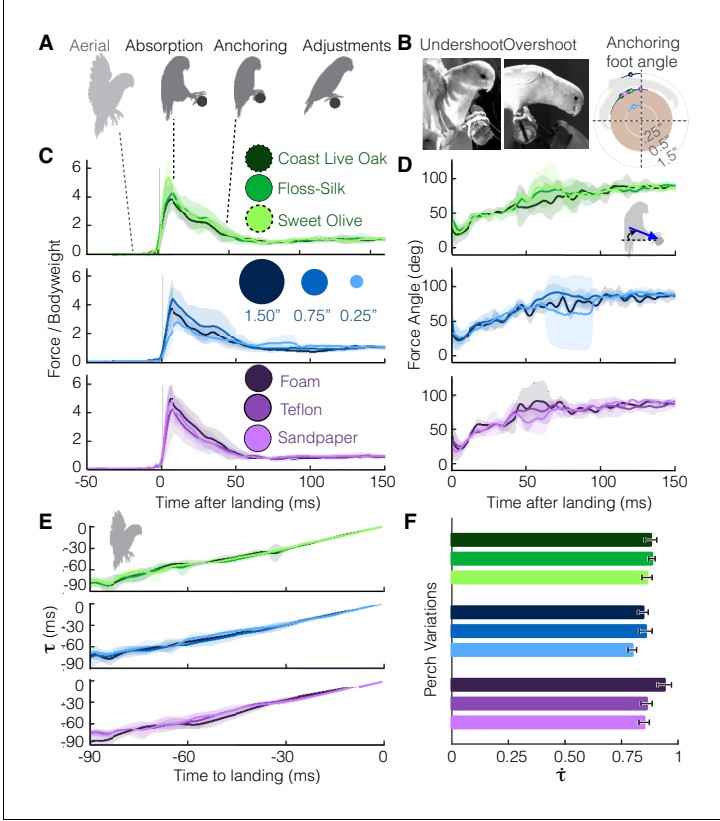

**Figure 2.** Despite differences in perches, parrotlets exhibit stereotyped landing behavior in their leg and wing dynamics. (A) Phases of landing (aerial, absorption, anchoring, adjustments) exhibited during each flight. (B) The parrotlets sometimes overshoot or undershoot the perch during landing. This may have increased the variance of the anchoring foot angles at which a bird stops slipping. The diagram illustrates the mean anchoring foot angles for the different surfaces along with their variance. Net force magnitudes (normalized by bodyweight) (C) and directions (D) also show variance, but average trends remain similar despite perch differences. The little variation in landing dynamics may be explained by how parrotlets control their landings; (E) the wings brake only enough to maintain a nearly constant change in rate of τ (estimated time to collision), (F) which is consistent with that of a controlled collision ($\dot{\tau} > 0.5$). (B-E) show mean ± SD for $N = 3$ birds, $n = 3$ flights per bird. (F) shows mean ± SEM for the same flights.

DOI: https://doi.org/10.7554/eLife.46415.005

indicate that the birds have not yet completed the braking process when they touch down, which likely leads to greater variance in their landing dynamics.

## Foot and claw kinematics

The parrotlets' stereotyped wing and leg behavior suggest that they compensate for perch differences with their feet and claws. However, the foot kinematics reveal a similarly stereotyped set of landing stages (*Figure 3A,B*) until the toes engage with the surface. The feet remain closed during flight ('resting'), and then about 100 ms before touchdown the foot spread angle begins to increase ('spreading,' 40 ms ± 8 ms duration) until the toes are fully outstretched ('open,' 21 ms ± 7 ms duration). This open stance may accommodate for some uncertainty in when and

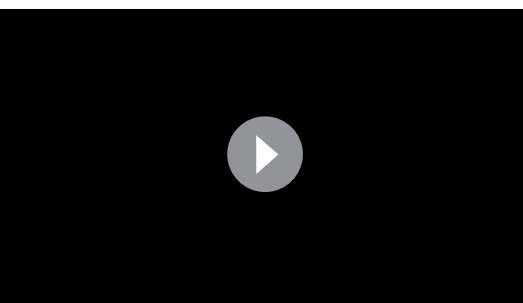

**Video 2.** Stereotyped landing phases (aerial, absorption, anchoring, adjustments) exhibited on each perch.

DOI: https://doi.org/10.7554/eLife.46415.006

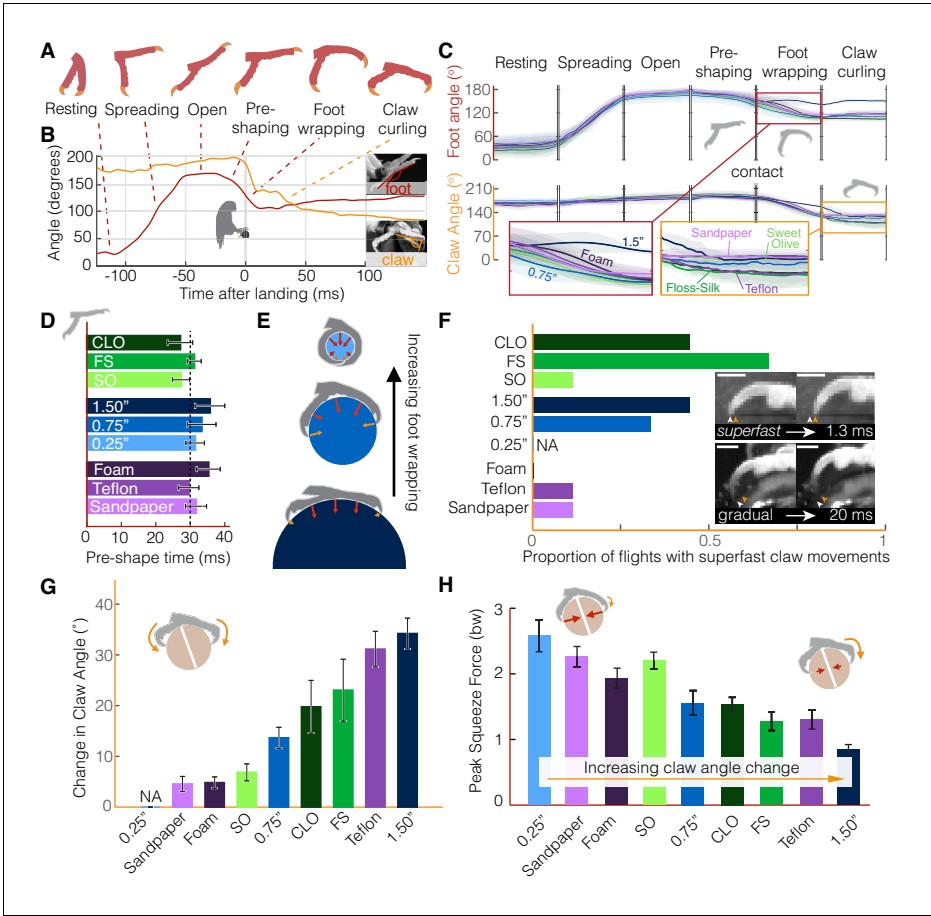

**Figure 3.** Parrotlets exhibit a stereotyped set of landing stages with their feet, yielding similar kinematics while in the air but surface-specific kinematics upon touchdown. (**A**) Avatars illustrate the phases of the feet during landing. The beginning and end of each phase are determined from (**B**) traces of the foot and claw angles (shown here for a representative flight). (**C**) Mean ± SD (*N* = 3 birds, *n* = 3 flights per bird) traces for each perch variation show how foot and claw kinematics are not surface dependent before touchdown but diverge after contact. Specifically, the insets illustrate how there is less foot wrapping on the large perch and more claw curling on the perches that are harder to grasp. Phase durations varied between flights, so the horizontal time axis is normalized for each phase. Traces for the 0.25" perch are not shown after contact, when the distal toepads and claws became obscured. (**D**) The time over which birds pre-shape their feet before touchdown suggests that parrotlets have predictive control resolution within approximately 30 ms. (**E**) After contact, the small 0.25" diameter perch enables more foot wrapping (>110% perch circumference) than the 0.75" perch (~50% perch circumference). The increased surface coverage enables birds to leverage normal forces in a larger range of directions around the perch. In contrast, the birds' feet reach less than 25% around the perch circumference on the larger 1.5" diameter perch. The diagrams show speculated gripping force vectors to illustrate how perch diameter may affect foot anchoring. (**F**) Claw curling motions can be gradual (on the order of 10–100 ms) or superfast (on the order of 1 ms), which suggests that potential energy stored in tendon elasticity may drive high speed claw movements. Superfast claw movements occur less frequently on rougher and smoother surfaces and did not occur on foam. Insets show superfast claw curling on the large diameter perch (upper) and gradual claw curling on the Teflon perch (lower). Arrows indicate the claw position before (white) and after (orange) claw tip motion and the scale bars are approximately 3.0 mm. (**G**) Parrotlets curl their claws most on the large diameter and Teflon covered perches and least on the sandpaper, Sweet Olive Tree, and foam-covered perches (NA: not applicable). These claw curling trends, taken together with (**H**) the peak squeeze forces from both feet, suggest that parrotlets curl their claws more on surfaces that are harder to squeeze. Bars show mean ± SEM for *N* = 3 birds, *n* = 3 flights per bird.
DOI: https://doi.org/10.7554/eLife.46415.007

The following figure supplements are available for figure 3:

**Figure supplement 1.** Superfast claw movements by bird.
DOI: https://doi.org/10.7554/eLife.46415.008

*Figure 3 continued on next page*

*Figure 3 continued*

**Figure supplement 2.** Time to minimum claw angle in the curl stage.
DOI: https://doi.org/10.7554/eLife.46415.009
**Figure supplement 3.** Claw curling in the front vs. rear claws.
DOI: https://doi.org/10.7554/eLife.46415.010

where the foot actually makes contact with the perch (as in *Figure 2B*). Immediately before contact, the toes start closing in ('pre-shaping,' 31 ms ± 10 ms duration), which suggests that the birds begin releasing the muscles holding their feet open when they sense that impact is more imminent. Pre-shaping the foot may also help the birds attain a secure grasp more quickly. After contact is made, the feet exhibit decreasing foot and claw angles as the toes finish closing around the perch ('foot wrapping,' 19 ms ± 7 ms duration). After conforming to the surface, the claw angle generally decreases during the 'claw curling' phase (185 ms ± 11 ms duration). The decrease in claw angle can result from the claw tip curling inwards towards a relatively stationary toe pad or the toe pad slipping outwards towards a relatively stationary claw tip. In some cases, both the toe pad and claw tip may move towards each other.

The different perch surfaces only affect the kinematics of the subsequent landing stages after perch contact (*Figure 3C*). Before contact, foot and claw kinematics remain consistent across all flights, even as the feet begin to close during pre-shaping. If the birds adjusted for perch differences while in the air, differences would be most likely to appear in this final stage before impact. Instead, the parrotlets consistently begin pre-shaping ~30 ms before touchdown (*Figure 3D*), suggesting that they have predictive control resolution within this time frame. Maintaining sufficiently large claw angles prior to this stage ensures that the claws are not damaged during the controlled collision with the perch. Only after contact do foot and claw kinematics begin to diverge. The extent of foot wrapping is governed by the diameter of the perch (*Figure 3E*). The parrotlets wrap their feet completely around the 0.25" diameter perch, which enables them to use normal forces from their toes to stabilize themselves. Although some claws still engage with the surface, there is no clear claw curling phase.

In contrast, the parrotlets exhibit foot wrapping and claw curling on all of the other surfaces. After contacting the perch, both the feet and claws can slip. As a result, the claw curl angle can change during the claw curling stage because the foot slips, the claw slips, or both slip, all of which can take place at varying speeds. Whereas the claw curling stage can last over 100 ms (based on visual observation, see Materials and methods), individual claw curling movements initiated when a claw slips off an asperity can take place as quickly as 1–2 ms (2–4% of a full wingbeat; *Figure 3F*, *Figure 3—figure supplements 1,2*). These superfast claw movements occur most frequently on intermediate surfaces that are neither extremely rough nor extremely smooth. This is likely because rough surfaces, such as Sweet Olive and sandpaper, have asperities that give sufficient friction so that there is little slipping (*Figure 3F,G*). On the other hand, smooth surfaces, such as Teflon, have very few usable asperities to slip off of or latch onto (*Figure 1A*, *Figure 5E*). Deformable surfaces, such as foam, do not have distinct asperities in the same way as the other surfaces, which explains why we did not observe any superfast claw movements on foam. The superfast claw movements are likely not governed by muscle contraction alone; muscle contractions in human hands are significantly slower (~50 ms) (*Buchthal and Schmalbruch, 1970*). Even the fastest vertebrate locomotory muscles recorded to date are still 5–10 times slower; the Etruscan shrew's extensor digitorum longus contracts in 11 ms (*Jürgens, 2002*) and the superfast pectoralis muscle of a hummingbird contracts in 8 ms (*Ingersoll and Lentink, 2018*). Therefore, the speed of the parrotlet claw movements probably rely on the release of energy stored in an elastic tendon (*McNeill Alexander, 2002*) and the low inertia of the claw. The superfast movements help ensure that the claws slip over timescales that are much shorter than the body dynamics timescales, which helps the bird remain more easily anchored on the perch.

In general, we find that the extent of claw curling depends on both perch diameter and surface properties (*Figure 3G*). Claw curling is the most pronounced on the 1.5" diameter perch (*Figure 3G*). Foot wrapping is much more limited on the large perch (*Figure 3C,E*), so birds must rely on shear forces from their toe pads and claws to generate sufficient stabilizing forces. The parrotlets tend to curl their claws less on perches that they can squeeze with more force, such as the

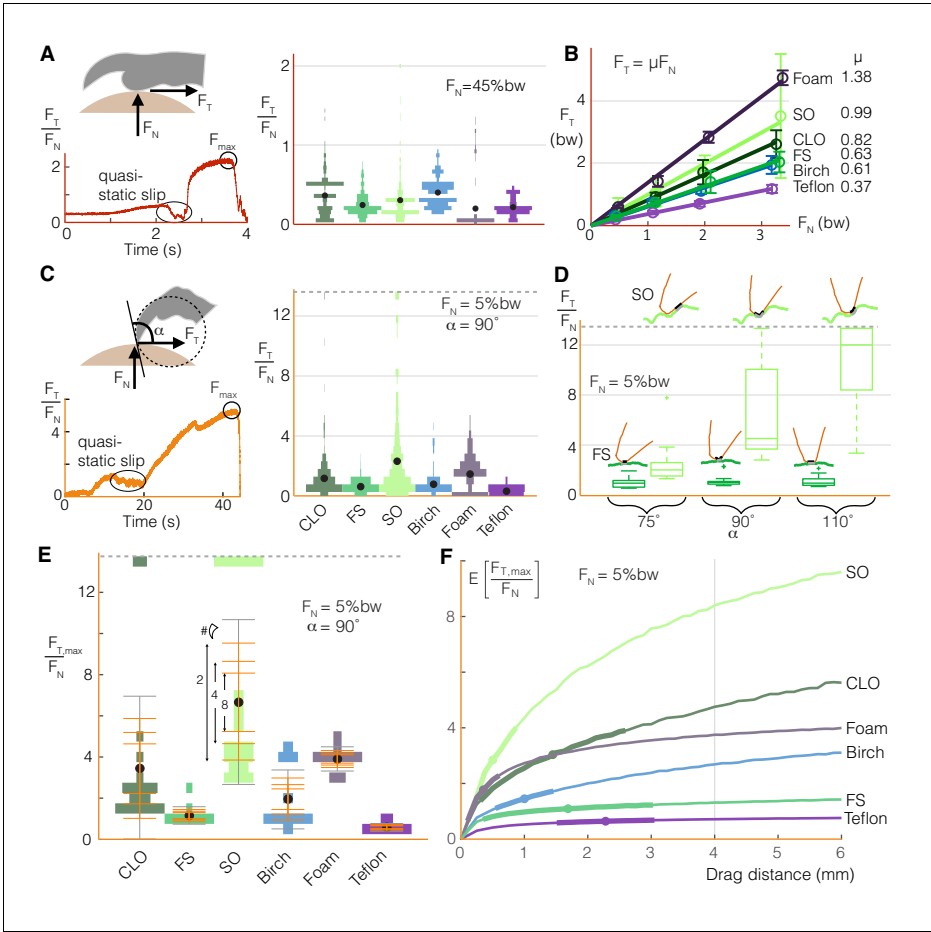

**Figure 4.** Birds can stably grasp different surfaces by modulating normal force on toe pads to generate predictable friction, and by dragging multiple claws to compensate for the variable nature of claw friction. (A) Toe pads slip and stick when dragged along a surface, as shown by the tangential force (friction) to normal force ratio, $F_T/F_N$, over time for a representative test. The violin plot illustrates the distribution of $F_T/F_N$ values immediately before slip events (bw = bodyweight, $d_{drag}$ = 5 mm) for Coast Live Oak (CLO), Floss Silk (FS), Sweet Olive (SO), birch, foam, and Teflon. Sandpaper, which yielded unnaturally high wear on claws and toe pads during pilot tests, was omitted from this testing. The number of peaks gives an estimate of the number of usable asperities, and the force before slip indicates how much friction those asperities provide. (B) Maximum friction values sustained at different normal forces demonstrate that toe pad friction is surface dependent and well described by a Coulomb friction model. Circles and error bars show mean ± SD for $n_{natural}$ = 10 (trials for natural surfaces), $n_{artificial}$ = 5 (trials for artificial surfaces). (C) Claws also slip and stick, as shown for a representative quasi-static drag test (see Materials and methods), but reach higher $F_T/F_N$ ratios with greater variance than toe pads ($n_{natural}$ = 20, $n_{artificial}$ = 10). (D) Max $F_T/F_N$ values, which approximate the friction a bird can expect from claw curling, increase with the angle between the claw and surface ($\alpha$) on the roughest natural surface (SO), but remain relatively constant on the smoothest (FS). Diagrams in D show claws at the microscopic level to illustrate how $\alpha$ (defined at the macroscopic level shown in C) affects interactions with surface asperities. (E) To compensate for variability in max claw friction, birds may leverage load sharing over multiple claws. Standard deviation bars based on sampling more claws from the single claw distributions, and averaging the force they produce, show that (as expected) the spread of the average claw force gets smaller as more claws contribute to generating force. Gray error bars show SD for the single claw distributions. Dashed lines in (C-E) show the max $F_T/F_N$ cutoff used to avoid claw damage during testing. (F) Birds can further improve available claw friction by increasing drag distance. The expected value of the maximum tangential to normal force ratio is denoted as $E[F_{T,max}/F_N]$. Dots and bolded segments of the curves denote estimated mean ± SD parrotlet claw drag distances, respectively, for each surface if the foot did not slip (see Materials and methods). The gray line indicates the estimated average drag length when landing, which was used for friction testing.

DOI: https://doi.org/10.7554/eLife.46415.011

rougher sandpaper and Sweet Olive perches. They curl their claws more on perches that they can squeeze less, such as the smooth Floss-Silk, the slippery Teflon, and the 1.5" diameter perch (*Figure 3H*). In short, the birds curl their claws more on perches that are harder to grasp.

## Foot-surface interactions

To understand mechanistically how varying squeeze force or claw curling enables the birds to adapt to different perch surfaces, we characterized foot-surface interactions using our surface testing setup (*Figure 1B*). Drag tests on the different surfaces reveal how toe pads (*Figure 4A*) and claws (*Figure 4C*) produce qualitatively similar shear force traces as they slip and stick. The maximum static shear forces correspond to the maximum force a bird could expect from a toe pad or claw at a given normal force. For toe pads, a Coulomb friction model is a suitable predictor of shear force as a function of normal force for all surfaces (*Figure 4B*). For claws, the friction ratios (tangential to normal force) can reach over eight times those generated by toe pads (*Figure 4C* vs. *Figure 4A*). These forces may also be affected by how the claw is oriented relative to the surface (*Figure 4D*); while the maximum static friction force stays relatively constant on the smoothest natural surface (Floss Silk Tree), it increases with claw angle on the roughest natural surface (Sweet Olive Tree). Even at a single angle, the variation of the maximum force ratios is much larger for claws than for toe pads (*Figure 4E*).

To compensate for the stochastic nature of claw engagement, birds can take advantage of load sharing and claw movement over the surface. Load sharing across multiple claws effectively narrows the spread of expected total claw forces (*Jiang et al., 2018*) (*Figure 4E*). Birds can also drag their claws over longer distances to improve the likelihood of hitting better asperities. This increases the expected value of the friction force (*Figure 4F*), which may explain why parrotlets curl their claws more on surfaces that are harder to squeeze (*Figure 3F*). The greatest benefits of dragging longer distances can be gained on surfaces that have fewer but larger asperities. However, the parrotlets tend to curl more on surfaces that give less expected benefit with claw travel (*Figure 4F*). This behavior suggests that the birds are not seeking out the best possible asperity when grasping. Rather, it suggests that they are instead curling their claws until they generate sufficient force to maintain a stable grasp or until they reach an upper limit. If the needed force is still not achieved by then, the bird may adjust its footing or take off.

## Modeling claw engagement

The greater variability in claw friction compared to toe pad friction may be explained by the local surface contact geometry. Toe pads, which are relatively soft, can conform to a surface. This allows them to distribute the frictional load across many contact points, leading to more consistent maximum friction forces. In contrast, claws, with a rigid structure and smaller contact area, rely on fitting into local surface geometries or hooking onto asperities. To quantify these geometric effects, we first characterized claw shape in the sagittal plane by width as a function of height from the claw tip (*Figure 5A*). The wide shape variation among parrotlet claws may result from the growth itself or from claw wear. High load drag tests (up to 3 times bodyweight, bw) caused approximately 900 µm of wear on a relatively sharp claw. Although quite high, the resulting claw geometry was still within the range of claw geometries that we measured in some of the duller, untested claws (*Figure 5A*, *Figure 5—figure supplement 1*). Drag tests at up to 25% bw caused no noticeable wear, showing how birds benefit markedly from limiting the normal force load on their claw.

In addition to using existing surface asperities, claws can also generate friction forces from surface deformation. To model this interaction, we combine measurements of claw penetration depth into different surfaces (*Figure 5B*) with the claw geometry. We find that the claw tip geometry can be modeled by a 50 µm radius sphere for loads up to 25% bw (*Figure 5C*). For example, the surface deformation from a normal force of 25% bw in Coast Live Oak can be approximated by a partial sphere with a 10 µm depth and 50 µm diameter. Therefore, although a 10 µm penetration depth is only about 0.2% of the outer arc length of a parrotlet claw (~5 mm), surface deformation can play an important role in maintaining a stable grasp, particularly on soft surfaces.

Next, to quantify the effect of claw size on claw engagement, we simulate claws with different tip radii dragging along measured surface profiles (*Figure 5D*, model adapted from *Asbeck et al., 2006*). The trajectory of the claw generates a *traced surface* (*Asbeck et al., 2006*; *Okamura, 2000*),

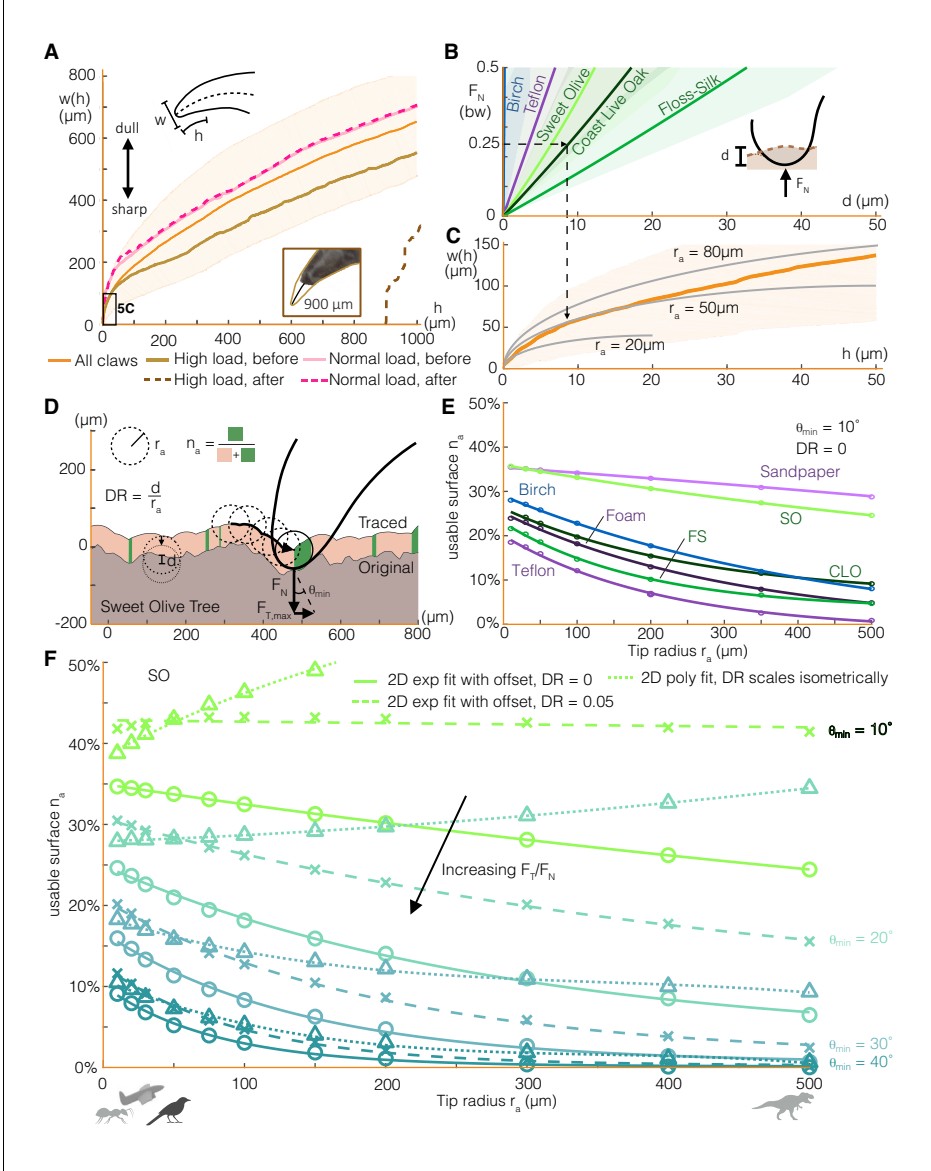

**Figure 5.** Claw-surface interactions can be modeled using a spherical claw tip, the size and depth of which relates nonlinearly to the number of usable surface asperities. (A) We characterize parrotlet claw geometry using width (w) as a function of height from the claw tip (h) in the sagittal plane. Testing within force limits expected during normal use (normal force $F_N$ = 25% bw) yielded no noticeable wear. In contrast, higher loads significantly altered claw geometry (see inset photo, $F_N$ = 300% bw). (B) The depth of surface deformation due to claw penetration is small compared to claw geometry (see *Surface deformation tests* in Materials and methods). This enables us to model claw-surface interaction geometry using a spherical claw tip (C). For example, the dashed arrows show that at $F_N$ = 25% bw on Coast Live Oak, the claw tip is well approximated by a tip radius $r_a$ of 50 μm. Bold lines and shading show mean ± SD for $n_{natural}$ = 20, $n_{artificial}$ = 10 tests in (B) and n = 32 claws in (A) and (C). (D) We simulate claws with different $r_a$ dragging along measured 2D surface profiles. Surface engagement is quantified by the usable surface ($n_a$), the proportion of the surface profile where static friction can be maintained. We also vary the depth ratio (DR, surface deformation depth divided by $r_a$) and the minimum contact angle ($\theta_{min}$) required to maintain static friction. (E) Rougher surfaces (*Figure 1A*) tend to have more usable surface (DR = 0, $\theta_{min}$ = 10°). The relationship between $r_a$ and $n_a$ for all surfaces is well approximated by an exponential fit with an offset. (F) Smaller $\theta_{min}$ amplify the effect of surface penetration, which increases usable surface, as shown by the growing discrepancy between the curves that account for surface penetration versus those that do not (DR = 0) for the Sweet Olive surface. Circles denote simulation results for DR = 0, and X's show DR = 0.05, which was estimated to be a reasonable value for a parrotlet landing (see Materials and methods). When depth ratio scales isometrically (triangles), the trends are flattened or even reverse for low $\theta_{min}$. Thus when steeper surface features are required,

*Figure 5 continued on next page*

*Figure 5 continued*

smaller animals and perching robots with lower claw tip radii tend to have more usable surface than larger animals (silhouettes show approximate tip radii scale for associated animals and a perching aerial robot), but the trend reverses when shallow asperities are sufficient.

DOI: https://doi.org/10.7554/eLife.46415.012

The following figure supplements are available for figure 5:

**Figure supplement 1.** Claw shape trends.

DOI: https://doi.org/10.7554/eLife.46415.013

**Figure supplement 2.** Surface deformation analysis.

DOI: https://doi.org/10.7554/eLife.46415.014

**Figure supplement 3.** Claw morphological measurement steps.

DOI: https://doi.org/10.7554/eLife.46415.015

**Figure supplement 4.** Claw-surface contact model.

DOI: https://doi.org/10.7554/eLife.46415.016

a smoothed version of the original profile that only comprises asperities that the claw can reach. We define the usable surface $n_a$ as the proportion of the traced surface with a slope greater than the minimum necessary for static friction ($\theta_{min}$). Specifically, $\theta_{min}$ is the angle formed between the vertical axis and the net force vector on the claw ($F_N + F_T$) when the tangential to normal force ratio is at its maximum. Starting with the case of no surface penetration, we find that rougher surfaces (*Figure 1A*) tend to have more usable surface (*Figure 5E*) for a fixed $\theta_{min}$. Still, because of the surface-specific distribution of asperity shapes and sizes, surfaces accommodate different claw sizes slightly differently. For example, despite exhibiting similar overall trends, birch offers more $n_a$ at smaller tip radius $r_a$ while Coast Live Oak has slightly more $n_a$ at larger $r_a$. An exponential fit with an offset describes the relationship between $n_a$ and $r_a$ well (*Figure 5E*). We might expect that as $r_a$ approaches infinity, $n_a$ should go to zero (no offset) because the traced surface flattens out. However, the offset can be understood by modeling the surface as a fractal; surface features with similar length scales to those shown here still exist at large $r_a$ (*Costa et al., 2000*; *Greenwood, 1992a*; *Greenwood, 1992b*). When surface penetration effects are added (*Figure 5F*), we find that the usable surface increases, with larger effects at lower $\theta_{min}$. If claw tip radius and mass both scale isometrically, and exerted forces scale with mass, then the depth ratio (*DR*, penetration depth divided by tip radius) will also scale isometrically. Although active muscle forces scale with length to the second power (*McMahon, 1984*), we assume that the forces involved in perching scale to the third power. This is because we expect these forces to be governed by bodyweight since perched animals must apply forces just large enough to maintain static equilibrium. Similar trends hold for constant and isometrically scaled depth ratios, *DR*, but the latter case yields trends that are flatter and eventually reverse for smaller surface slope requirements, $\theta_{min}$. Thus, smaller animals tend to have more usable surface than larger animals when steeper surface features are required, but the trend begins to reverse when shallow asperities provide sufficient friction.

## Integrated grasping model

We now zoom back out to see how the claws are integrated together with the feet and body in the perching behavior. We introduce a 2D rigid-body model of the parrotlet, in which aerodynamic and inertial forces and torques on the body must be matched by friction forces at the perch surface (*Figure 6A*). By leveraging our foot-surface contact mechanics data (*Figures 3–5*), we can now apply a constrained optimization to determine the bird's 3D 'wrench space' (*Ferrari and Canny, 1992*), the set of foot force and torque combinations that permit static grasping (see Materials and methods) (*Figure 6B*). Force plane cross sections show how torque twists and warps the stable boundary region. Thus, both the magnitude and direction of a bird's velocity when landing are important; if the bird's velocity vector is not directed at or near the perch center, the bird may slip. The stability region can be actively expanded by the bird through an increase in squeeze force, and is also enlarged when the surface texture itself presents more available surface friction (*Figure 6C*). Changing the angle of the foot relative to the perch rotates the wrench space (*Figure 6C*). The effect of perch size on the stability region may be less well-defined in nature. Smaller branches tend to enable greater squeeze forces (*Figure 3G*), and when the foot can fully

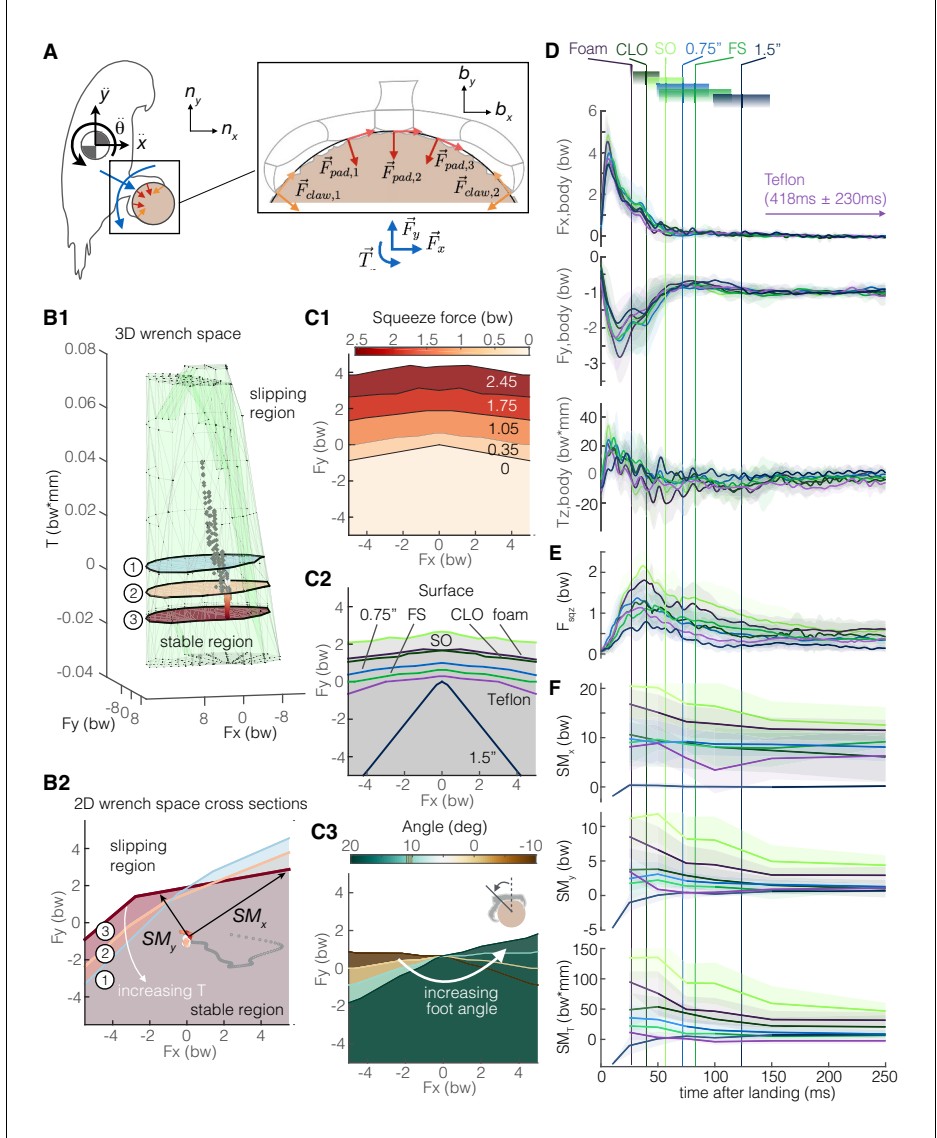

**Figure 6.** Adaptive grasping strategies that balance safety margin and required muscular squeeze force enable parrotlets to land reliably on variable surfaces. (A) We model the parrotlet as a 2D rigid body, where forces ($F_x$, $F_y$) and torques ($T_z$) on the body must be matched by friction forces between the perch surface and the feet and claws. (B1) By integrating our experimental data on the foot-surface contact mechanics (*Figures 3–5*), we use constrained optimization to generate a 3D wrench space in which birds can maintain a static grasp. (B2) 2D force plane cross-sections show how torque twists the wrench space. Forces from a single representative parrotlet flight are overlaid on the 3D (B1) and 2D (B2) plots, where colored points indicate static grasping and gray indicates slipping. We define safety margins ($SM_x$, $SM_y$, $SM_T$) in each direction of the foot reference frame ($b_x$, $b_y$, $b_z$) as the distance from the experimental data to the nearest static grasp region boundary in that direction ($SM_T$, not shown, extends out of plane). The stable region expands with more squeeze force (C1) and available friction on different surfaces (C2). (C3) The stable region also rotates with the foot's position on the perch. Aligning the stereotyped body force data (D), surface-dependent squeeze force data (E), and safety margins (F), we find that birds employ a landing strategy that balances safety margin and muscular squeeze force. The slow decay in the safety margins for most of the surfaces indicate that birds initially overcompensate after touchdown (prioritizing safety), and then slowly relax to equilibrium (prioritizing low muscular squeeze effort). Vertical lines and gradient blocks denote the average ± SD time at which birds stopped slipping on each surface. The safety margin (F) is plotted only when all claws were in contact with the surface.

DOI: https://doi.org/10.7554/eLife.46415.017

wrap around a perch, the ability to sustain forces depends only on muscular and structural limits. Thus, smaller branches should enable birds to withstand larger force disturbances. However, smaller branches can bend, which may make perching more difficult, and larger branches offer a larger area on which birds could potentially perch.

Finally, we apply this model to estimate how much more force and torque the birds apply during landing than would be necessary to maintain a static grasp. We define three safety margins, $SM_T$, $SM_y$, and $SM_x$, which correspond to how much more torque or force could be sustained without slip in the following directions, respectively: the axial torque direction ($b_z$), along a vector pointing from the center of the perch to the ankle ($b_y$), and the direction perpendicular to both $b_z$ and $b_y$ ($b_x$) (*Figure 6B*). For all surfaces, most of the body forces and torques are absorbed while the bird is still slipping (*Figure 6D*). Squeeze forces decay more slowly (*Figure 6E*) and appear to exert a greater influence over the safety margin trends (*Figure 6F*). Except for the large diameter perch flights, $SM_x$ values are much larger than $SM_y$ values and do not decay as much. This trend arises because pulling the foot in either the forward ($+b_x$) or aft ($-b_x$) direction increases normal and shear forces on one half of the foot, so $SM_x$ will remain relatively high even as a bird relaxes its grip. In contrast, pulling in the $b_y$ direction away from the perch decreases normal forces on the toes, so less force can be sustained in this direction without slipping. Safety margin peak values tend to be associated with both squeeze force and the quality of surface asperities for generating friction (*Figure 4B,E*). Perches with longer average slipping times, like the large diameter and Teflon perches, tend to have lower peak safety margins. These perches also do not consistently follow the same fast rise and slow decay trends in the safety margins of the other perches. This suggests that the ability to generate friction is the limiting factor for maintaining a stable grasp. However, we see positive safety margins even when birds are slipping, which suggests that some slipping may be intentional. Indeed, slipping does not always imply a lack of control; rather, it may help absorb body energy during landing, increase the chance of finding high-friction asperities, and help the bird slide into a more stable position at the top of the perch.

## Discussion

When landing, birds must balance safety margin and reducing muscle activation energy expenditure from squeezing. If birds prioritized safety during landing, we would expect consistently high grasp forces until all of the body energy had been absorbed. If they prioritized minimizing energy expenditure from squeezing, we would expect low, constant safety margins. In all landings, we find that the squeeze force and safety margins tend to peak later and decay more slowly than the body forces (*Figure 6D–F*). The reduction in squeeze force is likely aided by the high shear to normal force ratios that claws can achieve as soon as they lock onto an asperity (*Figure 4E*), enabling a stable grasp with relatively low normal forces. Reducing normal forces also has the added benefit of limiting claw wear that could result from high loading (*Figure 5A*). The later peak of the safety margins is likely a result of the delay between impact and wrapping the toes around the perch. These fast rise and slow decay safety margin trends on most of the surfaces indicate that the birds initially overcompensate – likely to help account for uncertainty due to both surface variability and landing variability (*Figure 2B*) from their controlled collisions (*Figure 2E*) – and then they slowly relax to equilibrium as they establish their grip. This 'overcompensation and then relax' strategy makes sense since landing safely is a critical requirement for preventing injury, while reducing muscular energy expenditure from squeezing improves energetics. Still, the safety margins are highly variable, likely due to variation between and within different surfaces, as well as variation in the bird's landing dynamics (orientation, velocity, etc.). It is still unclear how the birds use tactile feedback to adjust their grip during perching, which may drive how they decide to squeeze different surface textures.

To understand how the implications of these results may drive the grasping performance of other perching animals and robots, we must consider the effects of scale. In particular, the traced surface model (*Figure 5D–F*) indicates that small animals like insects (beetle tip radius = 8–10 µm; *Dai et al., 2002*) or even robots, which generally rely on sharp fish hooks for 'claws' (tip radius = 10–15 µm; *Lussier Desbiens and Cutkosky, 2010*; *Roderick et al., 2017b*), have more usable surface on vertical surfaces (high $F_T/F_N$) (see Materials and methods for details on the claw penetration depth testing). On the other hand, when more normal force is available (as when squeezing with large forces or on horizontal surfaces), larger animals, such as dinosaurs (*Manning et al., 2006*), have just as

many or more usable surface options (*Figure 5F*). Previous work (*Asbeck et al., 2006*; *Labonte et al., 2014*) reported a directly inverse relationship between asperities per unit length and tip radius at all length scales on rough building materials. The use of this metric was motivated by modeling the probability of hitting an asperity for a given travel distance along the surface. In this work, we chose to use usable surface rather than asperities per unit length because usable surface is independent of the spatial frequency of the surface. In other words, this method, unlike previous methods, assumes that many distinct usable asperities on a given section of surface are just as valuable to a grasping animal as a smaller number of larger asperities, provided that the usable surface in each case is equal and usable asperities will be encountered during the course of claw curling.

However, isometric scaling also suggests that structure strength (~length$^2$ since stress scales with area) diminishes with mass (~length$^3$), so larger animals may be unable to achieve high safety margins during grasping. On the other hand, they may not need to, because they generally deal with longer time scales and can tune control efforts more precisely (*Kovac, 2016*; *Roderick et al., 2017a*). Indeed, the safety margins employed by parrotlets (*Figure 6F*) are less than those used by beetles (*Voigt et al., 2017*) (~30), roughly on the same order as those used by snakes (*Byrnes and Jayne, 2014*; *Marvi and Hu, 2012*) (~5) and robots (*Estrada et al., 2014*) (~4), and generally larger than those used by humans (*Westling and Johansson, 1984*; *Cadoret and Smith, 1996*) (~1.5). While these safety margins are calculated in widely different contexts and do not reflect the different risks involved in grasp failure, they do suggest an association with scale, likely due to the structural and control differences.

In summary, parrotlets are able to use stereotyped aerial landing maneuvers across perch variations by adapting to perch differences with their feet and claws. During grasping, they use their toe pads to generate predictable friction forces, and their claws to generate higher but less predictable forces. Claws can be curled along the surface to increase the expected value of the friction force, and multiple claws used together reduce the variance of the average force generated per claw. Further, parrotlets employ an 'overcompensation then relax' grasping strategy which balances safety and reducing energy expenditure from squeezing. The separate, specialized functions of the legs and wings versus the feet and claws not only improve the reliability and efficacy of perching, but also simplify the aerodynamic and muscular control required.

Many of these landing and grasping insights can be applied more generally to both animals and robotic end-effectors. Our findings lend support to (1) the utility of modularity for simplifying control and improving individual subsystem performance, (2) load sharing, (3) taking advantage of regression to the mean when using toes and claws to handle the statistical variation of surfaces, and (4) grasp strategies that prioritize first safety then efficiency. These benefits and considerations are equally relevant for animals of all sizes and likely played key roles in shaping landing behavior across arboreal animals. Further, these grasping strategies likely also extend to takeoff maneuvers; stereotyped takeoff behaviors have been observed in previous work (*Provini and Abourachid, 2018*) as well, although more study is needed to similarly quantify takeoff dynamics over a range of complex surfaces.

The parrotlets' adaptive grasping strategies can also help inform the design of aerial robots, which are frequently equipped with small robotic arms to serve as landing 'legs' (*Roderick et al., 2017a*). Our results suggest that the mechanical design and control of aerial and perching components of multimodal robots can function independently. Regardless of perch surface, aerodynamic surfaces can be controlled the same way using feedforward control with optic flow to make controlled collisions. Meanwhile, the perching mechanism can provide robustness for different perches and surface irregularities by using avian-like grasping features. These features include foot wrapping with multiple joints, claw curling, and tactile feedback for adjustments. The traced surface and wrench space models presented here can give insights into the design tradeoffs between using more contact points, using claws and/or frictional pads, and determining a suitable balance of strength and sharpness in claws. Incorporating the perching capabilities of arboreal birds into aerial robots will greatly improve their versatility and utility in different environments.

# Materials and methods

## Perches

We fabricated nine different perches for this experiment: three 'natural', three birch dowel, and three 'artificial'. The three natural perches were cut directly from similar diameter branches of three trees on Stanford's campus: a Coast Live Oak (*Quercus agrifolia*) (15 mm dia.), a Floss-Silk tree (*Ceiba speciosa*) (16 mm dia.), and a Sweet Olive tree (*Osmanthus fragrans*) (17 mm dia.). To minimize moisture loss, the branch segments were kept in a sealed bag until testing, which was within three days of the branches being cut. The birch perches were cut from commercially available 0.25" (6 mm), 0.75" (19 mm), and 1.5" (38 mm) diameter birch dowels, all of which were colored black with a marker to match the color of the artificial perches (to control for color-based visiomotor bias). The three artificial perches were made from wrapping 0.125"-thick black (sponge neoprene) foam around a 0.5" (13 mm) diameter birch dowel, transparent PTFE (Teflon) tape around a 0.75"-dia. birch dowel (also colored black), and 80-grit black sandpaper around a 0.75"-dia birch dowel. A limitation of the split perch design was that the parrotlets could lodge a claw in the split when securing their grasp, but this happened infrequently (less than 25% of recorded flights). These events did not involve the claw tracked for kinematics, and we did not observe any noticeable effect on grasping dynamics, so we did not change our treatment of these cases.

## Parrotlet flight experiments

### Instrumented split perch

Each perch (see *Perches*) was split in half and mounted on two mechanically separated force sensors (ATI nano43 6-axis, 2 mN resolution) sampling at 3000 Hz (*Figure 1C*). To best measure squeeze forces from the feet, the split was oriented 20 degrees from the vertical, which was approximately where the parrotlets tended to place the center of their foot when taking off and landing during pilot tests. All forces were filtered using an eighth-order Butterworth filter with a 100 Hz cutoff frequency (~5x wingbeat frequency and below the natural frequencies of the perches, which were all at least 108 Hz).

### Birds and training

We trained three Pacific parrotlets (F. coelestis; 30 g, two male and one female, 20 Hz wingbeat frequency, 20 cm wingspan, *N* = 3) to fly between each instrumented split perch and a separate non-instrumented perch (0.625" diameter birch dowel), placed 50 cm apart. The perches were set at the same height and parallel to one another to keep flights as straight and level as possible. Each parrotlet was trained using habituation and positive reinforcement, wherein the bird was rewarded with millet seeds when it flew to the perch that the trainer pointed at using either a finger, a target stick or luring with a small piece of millet spray. One bird could not be trained to voluntarily land on foam, so landings for the foam perch were only recorded for two birds (*N* = 2). A different bird began to molt during the final day of experiments and was thus missing a few of its flight feathers for the artificial perch tests; however, this is unlikely to have affected its grasping or claw forces, and it still successfully landed on all of the perches without any noticeable differences in its flight behavior. All birds were trained to step voluntarily on and off a carry perch to be transported back to their housing between recordings (food and water provided ad libitum). Cages had enrichment, animals were not sacrificed for this study, and all training and experimental procedures were approved by Stanford's Administrative Panel on Laboratory Animal Care (APLAC-31426). Successful perch-to-perch flights were made by each parrotlet before recording any experimental data to avoid any confounding learning effects.

### Kinematics

Five high-speed cameras (four Phantom Miro M310s and one Phantom Miro LC310, 1280 × 800 resolution) filming at 3000 fps, synchronized with each other and the force sensors, were used to determine body, foot, and claw kinematics. Pilot studies indicated that 3000 fps was necessary for capturing claw kinematics. We defined superfast claw movements (*Figure 3F*) as when the claw moved by at least one pixel (approximately 0.3 mm) with respect to the foot center within 2 ms when starting from no perceptible motion. Cameras were calibrated using the DLT software

(*Hedrick, 2008*) with an average DLT error of less than 1%. To measure 3D body kinematics, the position of the bird's eye and tip of its tail were tracked in two camera views. To recover foot spread and claw angles, four points were tracked on the left foot, which was closest to the side view cameras. These four points were the center of foot, the tips of the most distal toe pads of the longest front and rear toes, and the claw tip of one of those toes. If the bird overshot the perch (determined by the visual extent and direction of slip), the rear claw was selected. If the bird undershot the perch, the front claw was selected. If there was no clear overshoot or undershoot, the front claw was selected. This selection criterion was used for two primary reasons: (1) The selected claw was expected to be the main source of stabilizing claw force, because toes are tendon-driven mechanisms that are specialized to curl with more force than they can exert to uncurl (*Bourbonnais et al., 1993*). (2) Despite camera angle optimization and filming with five cameras, the unselected claw occasionally became obscured by the perch during the overshoot or undershoot. The front claw was selected for 82% of the flights. There was no noticeable difference in the claw curl angle between the front and rear claw (*Figure 3—figure supplement 3*). These four points were tracked in a single 2D side view from when the bird's foot entered the camera view until either the foot stopped moving or the tracked foot left the perch after landing to adjust the grip. All tracking data was low-pass filtered (eight-order Butterworth, 100 Hz cutoff frequency).

The 0.25" diameter perch flights required slight modifications to this methodology because the toes could completely wrap around the perch. This foot wrapping enabled birds to rely on normal forces rather than shear forces to sustain external force loads. As a result, the claws did not consistently engage with the perch surface (in more than 40% of the flights, at least one claw was not engaged), and no claw curling phase was identified. Foot and claw angles were therefore only tracked during the aerial phase for this perch.

The Tau function (*Figure 2E*) was calculated by dividing the distance between the tracked position of the eye and the center of the instrumented perch by the time rate of change of that distance (velocity). A linear fit of the resulting Tau values ($R^2 = 0.96 \pm 0.04$) provided a reliable estimate of $\dot{\tau}$ (*Figure 2F*).

The foot and claw landing stages shown in *Figure 3A–C* were defined as follows:

- Resting: From the start of foot tracking (when the bird entered the camera frame) to when 10% of the peak foot spread angle was reached.
- Spreading: From 10% to 90% of the peak foot spread angle.
- Open: From 90% to 100% of the peak foot spread angle
- Pre-shaping: From the peak foot spread angle to the first point of contact ($t = 0$).
- Foot wrapping: From contact ($t = 0$) to when the tracked claw and associated toe make stable contact with the perch.
- Claw curling: From the end of foot wrapping to the end of tracking.

## Surface characterization

### 3D surface scanning

To characterize surface roughness, we 3D-scanned each perch (DAVID SLS-2 3D scanner; 19-50 μm, 20 μm average resolution). We used a best-fit cylinder to unwrap the surface to avoid warping surface features due to surface curvature when filtering. We then applied a low-pass filter with a cutoff wavelength of 1 mm (5% of the perch diameter, 20 times the claw tip radius). Next, we subtracted that filtered surface from the unwrapped point cloud. The resulting point cloud was then used to generate RMS values for each surface (*Figure 1A*), where $R_{RMS} = \sqrt{\frac{1}{n}\sum_{i=1}^{n} y_i^2}$ and $y$ is the local surface height. Additional roughness parameters can be found in the *Supplementary file 1A*. In order for the 3D scanner to detect surface features, dark and transparent surfaces (foam, sandpaper, Teflon, Coast Live Oak, and Floss Silk perches) were spray painted with flat white Krylon spray paint. We measured the weight of the non-natural surfaces before and after spraying them and, using the spray paint density, found the estimated average paint coating thickness to be < 30 μm. We estimate the coating thickness to be the same for the natural perches, though the thickness for these surfaces could not be measured directly because the natural perches would lose mass over time from evaporation.

## Friction drag tests

To characterize the parrotlet toe pad and claw friction properties on each surface (*Figure 4*), a split perch section from each surface was fixed to a 6-axis force sensor (ATI nano43, 2 mN resolution, 1000 Hz data collection frequency), which was mounted next to a linear slide (*Figure 1B*). A carriage holding a 600 mm carbon fiber beam that could freely pivot about its center was slowly pulled along the slide by an elastic cord connected to a motor. 3D-printed attachment pieces enabled the beam to hold a toe pad or claw (retrieved from the foot of a parrotlet cadaver, N = 1 for each treatment) at different angles on one end. The beam held precision weights (for varying the normal force applied to the toe pad or claw) on the other end. The use of the motor and linear slide enabled the claw and toe pad to be dragged consistently in the circumferential direction for all tests. To gain insight into the maximum friction that birds could expect to produce during natural behavior, drag distance was based on the upper bound of toe and claw travel observed during landings. The claw was dragged 4 mm, centered over the top of the perch. Because the drag distance was small relative to the perch diameter (such that the local surface angle that the claw encountered varied by only ±12°), the surface was locally relatively flat, which allowed us to assume a constant claw angle in our calculations. The toe pad was dragged 5 mm, starting from 1 mm before the top of the perch to avoid hitting asperities with portions of the toe pad not intended to contact the surface. As a result, the local surface angle that the toe pad encountered spanned 30°. Claws are very sensitive to hooking onto individual asperities (as compared to toe pads which have a larger spatial distribution of surface contacts), so to avoid missing asperities as a result of the system's mass/momentum, the motor pulled the claw quasi-statically. The length of the beam ensured that bumps encountered by the claw would not change the relative angle between the claw and the surface. Once the claw or toe pad passed the drag distance, the motor was stopped and then data collection finished. Tested locations on each surface were selected randomly (in mm increments) without replacement before each trial. For the toe pad, we conducted 10 trials on each natural surface and five trials on each non-natural surface (order of testing: Floss Silk (FS), Coast Live Oak (CLO), Sweet Olive (SO), Foam, Teflon, Birch, Sandpaper). Claw interactions were more stochastic, so we conducted 20 trials on each natural surface and 10 trials on each non-natural surface aside from sandpaper (order of testing: medium force 70° SO/FS, medium force 110° SO/FS, medium force 90° SO/FS/CLO/Teflon/Foam/Birch, low force 90° SO/FS, high force 90° FS/SO). We left out data on sandpaper friction due to the wear that it caused on both toe pads and claws. For all other surfaces, initial high load (~3 bodyweight, bw) testing led to substantial claw wear (*Figure 5A*), N = 1, so we repeated all tests at lower normal forces with another claw (~0.05 bw, what we estimate a single parrotlet toe/claw would commonly sustain during perching, N = 1). To avoid damaging the claw in these tests, dragging stopped when the measured force in the direction of pull reached 0.2 N (approximately 0.68 bw). The toe pad data was lowpass filtered at 5 Hz for finding the distribution of force peak values (*Figure 4A*), but no filtering was needed for finding the max values (*Figure 4B*). Data from the claw drag tests, which occurred over a longer time scale, were lowpass filtered at 1 Hz.

## Surface deformation tests

The surface deformation tests were conducted with the same setup as the *Friction drag tests*, except with a laser micrometer and no motor. The laser micrometer (Regular Reflective Displacement Sensor, Omron Z4M-N30V with sensor Z4M-S30V) was mounted above the perch surface to measure the depth of surface deformation during indentation tests with the parrotlet claw. The micrometer beam was directed at a ceramic plate fixed to the end of the carbon fiber beam, above the claw, to acquire depth change measurements. Using an OEMTools 25025 26 Blade Master Feeler Gauge, we measured a < 10 μm standard deviation of the accuracy. To isolate surface deformation from any deformation of the claw and claw mounting apparatus, we first calibrated the system by performing indentation tests on a steel plate. We then conducted 20 trials on each natural surface and 10 trials on each non-natural surface (order of testing: foam, Teflon, birch, sandpaper, Floss Silk, Coast Live Oak, Sweet Olive). The claw was moved a random distance (integer mm distance up to 10 mm) between each trial. During each trial, precision weights were removed from the other end of the beam to increase normal force on the claw. To limit the effect of sensor noise on our measurements, we tested at higher normal force levels (0.4 bw, one bw, 1.8 bw, and three bw) and applied a cubic fit constrained at the origin (*Figure 5—figure supplement 2B*) to estimate penetration at low force

levels (0–0.5 bw, see *Figure 4B*). Data for foam, which deformed over an order of magnitude more than any other surface, and sandpaper, which caused too much wear during friction tests, are included in the *Figure 5—figure supplement 2A*.

### Claw analysis and simulations

To understand how engaging more claws shrinks the variance of the average maximum friction force per claw during curling (*Figure 4E*), we created a Monte Carlo simulation using our measured single claw distributions (see *Friction Drag Tests*). For each surface, we averaged the result of randomly sampling once per engaged claw from the measured distribution. We then calculated the standard deviation of these averages over 10,000 trials. To determine the expected value of the maximum drag force as a function of drag distance (*Figure 4F*), we created a simulation using the measured distributions of all force peaks the claw encountered while dragging (*Figure 4C*). We chose to use a Monte Carlo simulation approach because no standard distribution fit the data very well. The number of force peaks is an approximation for the number of usable asperities the claw encountered, and the force at which the claw slipped indicates how good each asperity was. For each surface, we used averages of the number of asperities encountered in the trials to predict $N_a(d)$, the average number of asperities for a given drag distance $d$. We then ran 10,000 trials, during which we took the maximum of $N_a(d)$ samples from each force peak distribution across a range of $d$. Since $N_a$ was not always an integer value, we randomly selected either the nearest integer above or below $N_a$ during each trial such that the average number of samples over all trials equaled $N_a$. Finally, we averaged over all 10,000 trials to find the expected value of the maximum static friction force as a function of drag distance. The estimated parrotlet claw drag-lengths for each surface (*Figure 4F*) were determined by mapping the average claw curl angles (*Figure 3F*) to drag distances.

## Claw morphological measurements

We measured the external morphology of 32 claws from four parrotlets using a 5X microscope lens (Meiji Techno MX8000 Metallurgical Microscope). All claws were imaged in the sagittal plane at a resolution of 2560 × 1920 pixels. Photos of the claws were taken with a QImaging MicroPublisher 5.0 RTV (01-MP5.0-RTV-R-CLR-10 Color, RTV 10 Bit) camera and then post-processed in Matlab to determine the width of the claw as a function of the height from the claw tip. The specific steps of the algorithm are described in detail in *Figure 5—figure supplement 3*.

## Models

### Claw surface interaction model

We simulated a claw dragging along each surface using a 2D 'traced surface' method (*Asbeck et al., 2006*; *Okamura, 2000*) assuming each claw moved approximately in 2D. We used MATLAB's polybuffer function to compute the 'traced surface,' the surface created from the trajectory of the center of the claw tip moving over the original surface profile (*Figure 5D*). The claw tip was modeled as a 2D circle of varying radius, and the surface profiles were those measured with 3D scanning. We use the 'traced surface' to determine the amount of usable surface, the proportion of the surface with a slope greater than the minimum necessary for static friction ($\theta_{min}$) (*Figure 5D,E*). To calculate that proportion, we compare the slope of the traced surface over 1 μm distances to $\theta_{min}$. We adjusted $\theta_{min}$ to account for surface penetration using a classical contact mechanics model (*Popov, 2010*), in which the interaction is modeled as a circle (the claw tip) contacting an elastic half space (the surface). Thus, in modeling the claw-surface interaction, we assume the surface is a uniform elastic medium (no plastic deformation) that is infinitely deep and wide. To determine how far the claw penetrated into the surface, we first assumed a constant depth ratio $DR$ (surface deformation depth divided by tip radius) of 0.05 based on our surface deformation tests (*Figure 5B,C*); a tip radius of 50 μm at a normal force of 10% bw yields a deformation of ~2.5 μm, so $DR = 2.5$ μm / 50 μm = 0.05. We next assume that the depth ratio scales isometrically as tip radius changes. We then found the slope from the end of the claw tip to the highest contact point with the surface (which is deeper than the original surface height without claw interaction). The derivation of the equations we used to determine the slope are as follows:

To model the effect of surface deformation on the claw's usable surface, we modeled the claw-surface interaction as a rigid circle (the claw tip) contacting an elastic half-space (the surface), as

shown in *Figure 5—figure supplement 4*. We then found the slope ($m$) from the end of the claw tip ($x_{S1}$, $y_{S1}$) to the highest contact point with the surface ($x_{S2}$, $y_{S2}$):

$$m = \frac{y_{S2} - y_{S1}}{x_{S2} - x_{S1}}$$

As diagrammed in *Figure 5—figure supplement 4*, for a claw tip with radius $R$, $x_{S1} = 0$, $y_{S1} = -R$, $x_{S2} = a$, and $R^2 = x_{S2}^2 + y_{S2}^2$. We note that in the diagram $y_{S2}$ is negative, which gives,

$$y_{S2} = -\sqrt{R^2 - x_{S2}^2} = -\sqrt{R^2 - a^2} ,$$

Thus, we can simplify the equation for the slope as,

$$m = \frac{y_{S2} - y_{S1}}{x_{S2} - x_{S1}} = \frac{-\sqrt{R^2 - a^2} - (-R)}{a - 0} = \frac{-\sqrt{R^2 - a^2} + R}{a} .$$

To avoid confusion with the notation, we denote the depth ratio, DR, here as $D_R$,

$$D_R = \frac{d}{R} .$$

From Hertzian Contact Theory (*Popov, 2010*),

$$a = \sqrt{Rd} .$$

Simplifying the above two equations,

$$d = \frac{a^2}{R} ,$$

$$D_R = \frac{\frac{a^2}{R}}{R} = \frac{a^2}{R^2} ,$$

$$a = \sqrt{R^2 D_R} .$$

We now again simplify the equation for the slope,

$$m = \frac{-\sqrt{R^2 - a^2} + R}{a} = \frac{-\sqrt{R^2 - \sqrt{R^2 D_R}^2} + R}{\sqrt{R^2 D_R}} = \frac{R - \sqrt{R^2 - R^2 D_R}}{\sqrt{R^2 D_R}} = \frac{R - R\sqrt{1 - D_R}}{R\sqrt{D_R}} .$$

Finally, we thus find the following relationship for the slope $m$,

$$m = \frac{1 - \sqrt{1 - D_R}}{\sqrt{D_R}}$$

To adjust the required minimum angle necessary for maintaining static friction ($\theta_{min}$), we subtract the angle corresponding to the slope $m$ from the original minimum static friction angle $\theta_{min}$.

## Stable grasp model

We model the parrotlet as a 2D rigid body in a Newtonian reference frame n, where forces and torques on the body must be matched by friction forces between the perch surface and the feet and claws at every time step. In this model, the bird's two feet are represented by a single planar (2D) foot. The forces that would be exerted on the bird's two feet are therefore grouped to act on the single model foot. Similarly, the planar model foot groups the bird's four front toes and four rear toes into one front model toe and one rear model toe, respectively. Therefore, the model does not account for lateral forces out of the plane or torques in the plane. In addition, the model is only valid once the bird's feet have fully wrapped around the surface and the claws and toe pads have made contact. Mathematically, infinitely many contact points with the surface could exist along a bird's toes, but for computational feasibility and modeling simplicity (using the principle of parsimony), we

model the foot as having 5 contact points: 3 toe pad contacts (P1, P2, and P3) and 2 claw contacts (C1 and C2). One toe pad contact is located at the center of the foot. The other contacts are reflected symmetrically over the line connecting the center of the foot and the center of the perch (*Figure 6A*). The location of these contacts is a varied parameter that depends on perch diameter (discussed below). Origin O is located at the center of the perch. By varying the required $F_x$, $F_y$, and $T_z$ and integrating our experimental data on foot-surface contact mechanics, we use constrained optimization to determine the 3D wrench space (*Ferrari and Canny, 1992*) in which birds can maintain a stable grasp (*Figure 6B*). In the equations below, $\vec{F_x}$, $\vec{F_y}$, and $\vec{T_z}$ represent the external forces on the foot that the foot must counteract. Force and torque balances set 3 equality constraints:

Equality constraint 1: $b_x \cdot \left(\vec{F}_{claw,1} + \vec{F}_{claw,2} + \vec{F}_{pad,1} + \vec{F}_{pad,2} + \vec{F}_{pad,3}\right) = b_x \cdot \left(\vec{F_x} + \vec{F_y}\right) = F_x$

Equality constraint 2: $b_y \cdot \left(\vec{F}_{claw,1} + \vec{F}_{claw,2} + \vec{F}_{pad,1} + \vec{F}_{pad,2} + \vec{F}_{pad,3}\right) = b_y \cdot \left(\vec{F_x} + \vec{F_y}\right) = F_y$

Equality constraint 3:

$$b_z \cdot \left(\vec{r}^{C1/O} \times \vec{F}_{claw,1} + \vec{r}^{C2/O} \times \vec{F}_{claw,2} + \vec{r}^{P1/O} \times \vec{F}_{pad,1} + \vec{r}^{P2/O} \times \vec{F}_{pad,2} + \vec{r}^{P3/O} \times \vec{F}_{pad,3}\right) = b_z \cdot \vec{T_z} = T_z$$

Squeeze (internal) forces (*Figure 3G*) set two equality constraints (one on each side of the foot) that act only in the x direction:

Equality constraint 4: $b_x \cdot (\vec{F}_{claw,1} + \vec{F}_{pad,1} + \vec{F}_{pad,2}) = F_{sqz} + \max(F_x, 0)$

Equality constraint 5: $b_x \cdot (\vec{F}_{claw,2} + \vec{F}_{pad,2} + \vec{F}_{pad,3}) = -F_{sqz} + \min(F_x, 0)$

Shear forces in both directions are bounded by the available friction that we found for each surface (*Figure 4B*, *Figure 4E*), as is further described below. For the *i*th contact on a surface with friction coefficient $\mu_{surface}$,

Inequality constraints (1-5): $0 \leq \|\vec{F}_{i,shear}\| \leq \mu_{surface} \|\vec{F}_{i,normal}\|$

Normal forces on the claws were bounded by the maximum normal force used in the first set of friction drag tests (three bw), as we found that the claw sustained substantial wear from this load (*Figure 5A*). This value of 3 bw was also approximately the maximum squeeze force we measured (*Figure 3G*).

Inequality constraints 6: $\left\|\vec{F}_{claw,1,normal}\right\| \leq 3\ bw$

Inequality constraints 7: $\left\|\vec{F}_{claw,2,normal}\right\| \leq 3\ bw$

The constrained optimization, implemented through MATLAB's fmincon function, seeks to minimize normal forces from the toe pads and claws while satisfying these constraints. The constants used by the model are as follows:

- Location of foot-surface contacts: The locations of the five contact points (*Figure 6A*) were based on toe and claw contact points observed in our videos. One toe pad contact was in the center of the foot. For the 0.75" perches, the claw contacts were ±85° from the center of the foot. In reality, there are many contact points that exist along each toe, between the center of the foot and the claws, and these points can shift as the bird adjusts its grip. To keep computation time reasonable for our model, we grouped the remaining contact points into two toe pad contacts at ±45° from the center of the foot, which is roughly where we observed the most consistent toe-surface contact. For the 1.5" perch, the two claw contacts were 45° from the foot center and the other two toe contacts were 20° from the foot center.
- Surface diameter: 0.75" or 1.5"; the smallest perch is excluded as clarified below.
- Friction relationships: Friction was modeled using Coulomb friction, where the friction coefficient describes the maximum possible shear to normal force ratio. The friction coefficients were established using the data from the *Friction Drag Tests* (*Figure 4B,E*). For the claws, we used average friction coefficients from the 5% bw normal force tests. Although claw friction can vary substantially on real surfaces, load sharing across four claws on each foot would bring the average force per claw closer to the average of the single claw distribution (*Figure 4E*), which is most parsimonious.
- Landing angle: The orientation of the foot relative to the perch was based on the average initial anchoring foot angle measured for each surface (*Figure 2B*).
- Bodyweight: Equilibrium vertical force measured by the force sensors (*Figure 1D*).

The smallest diameter perch is not well described by this model because, on this perch, the bird is limited primarily by muscular and structural constraints instead of surface friction. The results from the small diameter model were therefore excluded from the safety margin plots. Finally, we note that while one might expect a negative safety margin during the slipping stage, safety margins may actually be positive if a bird intentionally delays static grasping (even when it is achievable) until a more stable position is reached.

## Acknowledgements

We thank Y Wang for her help in this study, Arul Suresh for his valuable insights into experimental design and analysis, and members of the BDML and Lentink labs for their support.

## Additional information

### Competing interests

David Lentink: Reviewing editor, *eLife*. The other authors declare that no competing interests exist.

### Funding

| Funder | Grant reference number | Author |
|---|---|---|
| National Science Foundation | CAREER Award 1552419 | David Lentink |
| Air Force Office of Scientific Research | DESI FA9550-18-1-0525 | Mark R. Cutkosky David Lentink |
| National Science Foundation | Graduate Research Fellowship | William R. T. Roderick |
| Department of Mechanical Engineering, Stanford University | Graduate Fellowship | Diana D. Chin |
| U.S. Department of Defense | National Defense Science and Engineering Graduate Fellowship | Diana D. Chin |

The funders had no role in study design, data collection and interpretation, or the decision to submit the work for publication.

### Author contributions

William RT Roderick, Conceptualization, Resources, Data curation, Software, Formal analysis, Funding acquisition, Validation, Investigation, Visualization, Methodology, Writing—original draft, Project administration, Writing—review and editing, WRTR contributed to collecting and analyzing data from the parrotlet experiments, interpreting the findings, and drafting the manuscript. WRTR also collected and analyzed surface friction and deformation data and developed the corresponding models; Diana D Chin, Conceptualization, Resources, Data curation, Software, Formal analysis, Funding acquisition, Validation, Investigation, Visualization, Methodology, Writing—original draft, Project administration, Writing—review and editing, DDC contributed to collecting and analyzing data from the parrotlet experiments, interpreting the findings, and drafting the manuscript; Mark R Cutkosky, Conceptualization, Resources, Supervision, Funding acquisition, Methodology, MRC contributed to interpreting findings; David Lentink, Conceptualization, Resources, Supervision, Funding acquisition, Visualization, Methodology, Project administration, Writing—review and editing, DL contributed to interpreting findings and editing the manuscript. DL also oversaw the project

### Author ORCIDs

William RT Roderick https://orcid.org/0000-0002-6998-1009
Diana D Chin https://orcid.org/0000-0002-3015-7645
Mark R Cutkosky https://orcid.org/0000-0003-4730-0900

### Ethics

Animal experimentation: All birds were trained to step voluntarily on and off a carry perch to be transported back to their housing between recordings (food and water provided ad libitum; cages had enrichment, animals were not sacrificed for this study, and all training and experimental procedures were approved by Stanford's Administrative Panel on Laboratory Animal Care, APLAC-31426). Successful perch-to-perch flights were made by each parrotlet before recording any experimental data to avoid any confounding learning effects.

### Decision letter and Author response

Decision letter https://doi.org/10.7554/eLife.46415.023
Author response https://doi.org/10.7554/eLife.46415.024

## Additional files

### Supplementary files

• Supplementary file 1. Supplementary tables. (**A**) Surface roughness parameters. Roughness parameters are based on processing 3D scans of each surface (see Materials and methods). (**B**) Individual foot preferences and adjustments during landing. Just as humans exhibit hand preference, birds appear to employ a dominant foot when carrying out different tasks. A previous study considered foot dominance in the context of manipulating food and found that approximately 50% of the parrots studied were left-footed, 25% were right-footed, and the remaining were ambidextrous (*Magat and Brown, 2009*). In our study, based on which foot contacted the perch first, we found that one bird was right footed (100% of flights) while the other two were left footed (83% and 85% of flights). We also looked at how the birds adjusted their feet after making initial contact. The first foot to move was the dominant foot for two individuals, but not for the third. In addition, we found that the first foot adjustment was primarily in the forward direction (92%, 83%, and 72%). (**C**) Parrotlet foot adjustments on each perch. Parrotlets made, on average, about two adjustments after landing. The birds adjusted their feet the most on teflon (2.78 adjustments), and the least on the 0.25" diameter birch dowel (1.33 adjustments). (**D**) Foot kinematics parameters. The average foot spread angle while in the resting stage was approximately 36°. When approaching the perch, the birds began to open their feet approximately 90 ms before making contact with the surface. During this time, the foot angle reached a maximum of 172° on average, and this peak typically occurred approximately 31 ms before making contact with the surface. While pre-shaping, the foot angle decreased approximately 24° on average. Before making contact with the perch, the average claw angle was 177°. After having made contact, while wrapping the perch, the foot angle dropped another 30° on average. The claw angle, on the other hand, dropped an average of 42° during this stage. The claw reached a maximum curl angle typically 85 ms after contacting the perch, though with a high standard deviation of 80 ms.
DOI: https://doi.org/10.7554/eLife.46415.018

• Transparent reporting form
DOI: https://doi.org/10.7554/eLife.46415.019

### Data availability

All data files are available on Dryad, https://doi.org/10.5061/dryad.89rr53h.

The following dataset was generated:

| Author(s) | Year | Dataset title | Dataset URL | Database and Identifier |
|---|---|---|---|---|
| Roderick WRT, Chin DD, Cutkosky MR, Lentink D | 2019 | Data from: Birds land reliably on complex surfaces by adapting their foot-surface interactions upon contact | https://doi.org/10.5061/dryad.89rr53h | Dryad Digital Repository, 10.5061/dryad.89rr53h |

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
