## [Decision Letter]

Thank you for submitting your article "Birds land reliably on complex surfaces by adapting their foot-surface interactions upon contact" for consideration by *eLife*. Your article has been reviewed by two peer reviewers, and the evaluation has been overseen by a guest Reviewing Editor and Diethard Tautz as the Senior Editor. The following individual involved in review of your submission has agreed to reveal his identity: Andrew Biewener (Reviewer #2).

The reviewers have no major critical comments. Reviewer #2 lists a number of specific comments that can be considered as minor, and it should be easy to address them.

*Reviewer #1:*

The experimental results presented in this study are interesting. The theory analysis uses a very simple approach and I am not sure how useful this is but it is also not clear to me how to improve it. I recommend publication as it is!

*Reviewer #2:*

The authors report detailed, careful measurements and analyses based on well-designed experiments to examine how the feet (claws and toe pads) of Pacific Parrotlets grip natural and artificial perches of varying surface roughness, stiffness and diameter to achieve stable grips upon landing. Their findings provide novel insight for understanding the robust and adept ability of many species of birds to land on arboreal perches. The paper is generally clear and well-written. However, the amount of information that is condensed into several of the figures makes certain aspects of what is shown difficult to follow and to interpret clearly. Specific aspects are detailed below. Nevertheless, I enjoyed reading the paper and following the authors' analyses and findings. With generally minor revisions, this will make for an important contribution, of interest both to engineers and biologists.

Specific comments:

"This explains how many animals and robots can grasp complex surfaces reliably." In the case of aerial robots, I think "may" not "can" is most appropriate. I don't know of any aerial robot that approaches the ability of birds to land and grasp 'complex surfaces'.

Figure 2B is hard to interpret; I found that the information in the legend and the figure panel was insufficient. The results for this panel could be better explained.

"the majority of the claw curling can take place as quickly as 4 ms (8-9% of a full wingbeat) if a claw slips off of an asperity. If the claw latches onto multiple asperities as it curls, then the curling becomes much more gradual." What is the evidence (data) for this?

"the relatively fragile claws are…" What evidence exists that claws are 'fragile'? I am not convinced that claws are "fragile" structures. Is there evidence that birds may frequently fracture their claws on landing? Although, as the authors show, dragging them over unnatural sandpaper wears down the tip, I know of no evidence for their being 'fragile'. I suggest deleting this to make for a more objective statement.

Again, Figure 3C insets are hard to follow/interpret.

Figure 3E – clarify if arrows are speculated gripping vectors – only total grip force can be measured with setup used. Also, the 1.5" diameter perch shows foot wrap far less than 90% circumference (but <90% is stated in legend). The figure shows this to be <25%, at most. (Shear forces also presumed not measured).

Figure 4C clarify that time axis is correct (nearly order of magnitude > 4A time scale). 20-40 seconds doesn't make sense unless the claw was dragged VERY slowly.

Figure 4D is unclear/too fine scale to see clearly in figure. Also, I am not convinced that the angle of claw engagement can be accurately discriminated between 75, 90 and 110 deg for the smoothest surface, as shown. To my eye, the 110 deg panel shows an angle closer to 0 deg (horizontal).

Figure 4E OK, but this is an unsurprising result.

Figure 4F notation for Mean + SD is not very clear. what does "E" refer to in this panel (by ratio of tangential to normal force)?

Figure 5 What does "highly non-linearly" mean? How is this evaluated? Need an objective measure for this characterization. Suggest simply "non-linearly".

Unclear how 5B data are established.

θmin is not clearly defined or illustrated in 5D – reference should be after 'angle', not after 'friction'; and "As shown for the Sweet Olive surface, smaller θmin amplify the effect of surface penetration…" is unclear to me how this amplifies surface penetration effect.

"Specifically, θmin is the angle formed between a given normal force (FN) and the corresponding maximum tangential force vector (FT)." How can the angle between FN and FT not be 90 deg?

Figure 5F: silhouettes aren't mentioned in legend, or clear why they are shown.

Figure 6B and C are unclear to this reviewer. 6D and E are the most clear and relevant to this reviewer.

Unclear why "structure strength (~length^2^)" Isn't this an area?

Also, it would help to reference methods here, re: how claw penetration depths were measured on different surfaces.

Subsection “Friction drag tests”: This is also unclear: For friction drag tests, were claws and toe pad pulled along beam length or in circumferential direction? If the former this would presumably have different asperities/roughness measures. If the latter, explain more clearly and how/why surface curvature does not affect the analysis (were curvature effects removed beforehand?)

Otherwise, the authors are to be congratulated on impressive set of measurements, analysis and findings.

---

## [Author Response]

Reviewer #2:

[…] Specific comments:"This explains how many animals and robots can grasp complex surfaces reliably." In the case of aerial robots, I think "may" not "can" is most appropriate. I don't know of any aerial robot that approaches the ability of birds to land and grasp 'complex surfaces'.

Complied; we changed “can” to “may”.

Figure 2B is hard to interpret; I found that the information in the legend and the figure panel was insufficient. The results for this panel could be better explained.

Complied; we agree (thank you for catching this) and now further explain Figure 2B in the caption by adding, “The diagram illustrates the mean anchoring foot angles for the different surfaces along with their variance.”

"the majority of the claw curling can take place as quickly as 4 ms (8-9% of a full wingbeat) if a claw slips off of an asperity. If the claw latches onto multiple asperities as it curls, then the curling becomes much more gradual." What is the evidence (data) for this?

Complied and clarified; like the reviewer, we were surprised by this unusually fast behavior and agree it is worthwhile to better show how it is supported by the underlying data.We addressed this by showing the underlying data in a new figure panel (Figure 3F). We have also added two new figure supplements (Figure 3—figure supplement 2, 3) that further illustrate the variability of claw curling. These figures helped us quantify and clarify the statement in the main text, which now reads:

“In contrast, the parrotlets exhibit foot wrapping and claw curling on all of the other surfaces. […] The superfast movements help ensure that the claws slip over timescales that are much shorter than the body dynamics timescales, which helps the bird remain more easily anchored on the perch.”

"the relatively fragile claws are…" What evidence exists that claws are 'fragile'? I am not convinced that claws are "fragile" structures. Is there evidence that birds may frequently fracture their claws on landing? Although, as the authors show, dragging them over unnatural sandpaper wears down the tip, I know of no evidence for their being 'fragile'. I suggest deleting this to make for a more objective statement.

Complied; we removed the word “fragile.” We had initially used that term because we had been comparing them to the steel fishing hook claws used by many perching robots, but we agree these claws are not fragile at all (in fact one of the senior authors had commented on it, but it accidentally didn’t get revised in our earlier manuscript).

Again, Figure 3C insets are hard to follow/interpret.

Clarified; We clarified the caption for Figure 3C by adding: “Specifically, the insets illustrate how there is less foot wrapping on the large perch and more claw curling on the perches that are harder to grasp.”

Figure 3E – clarify if arrows are speculated gripping vectors – only total grip force can be measured with setup used.

Complied; we now clarify in the caption for 3E that the arrows indicate speculated gripping forces as follows: “The diagrams show speculated gripping force vectors to illustrate how perch diameter may affect foot anchoring.”

Also, the 1.5" diameter perch shows foot wrap far less than 90% circumference (but <90% is stated in legend). The figure shows this to be <25%, at most. (Shear forces also presumed not measured).

Complied, thank you for catching this. We corrected “<90%” to “<25%” (we had initially intended to write <90˚). We also removed the mention of shear forces in that sentence. The sentence now reads: “In contrast, the birds’ feet reach less than 25% around the perch circumference on the larger 1.5” diameter perch.”

Figure 4C clarify that time axis is correct (nearly order of magnitude > 4A time scale). 20-40 seconds doesn't make sense unless the claw was dragged VERY slowly.

Complied; we now clarify that the drag test was quasi-static and added a reference to the Materials and methods. In the Materials and methods under “Friction drag tests”, we clarified as follows: “Claws are very sensitive to hooking onto individual asperities (as compared to toe pads which have a larger spatial distribution of surface contacts), so to avoid missing asperities as a result of the system’s mass/momentum, the motor pulled the claw quasi-statically.”

Figure 4D is unclear/too fine scale to see clearly in figure. Also, I am not convinced that the angle of claw engagement can be accurately discriminated between 75, 90 and 110 deg for the smoothest surface, as shown. To my eye, the 110 deg panel shows an angle closer to 0 deg (horizontal).

Clarified; The angle of claw engagement referred to in Figure 4D is defined at the macroscopic level (shown in 4C), rather than the microscale level shown in the 4D diagrams. We added a clarifying statement in the caption for Figure 4 to explain this: “Diagrams in D show claws at the microscopic level to illustrate how 𝛼 (defined at the macroscopic level shown in C) affects interactions with surface asperities.”

Figure 4E OK, but this is an unsurprising result.

Complied; We agree and have adjusted the phrasing in the caption for Figure 4E to clarify that the result was expected: “Standard deviation bars based on sampling more claws from the single claw distributions, and averaging the force they produce, show that (as expected) the spread of the average claw force gets smaller as more claws contribute to generating force.”

Figure 4F notation for Mean + SD is not very clear. what does "E" refer to in this panel (by ratio of tangential to normal force)?

Clarified; We improved our notation in Figure 4F to more clearly show mean and standard deviation with dots and bolded segments of the data curves, respectively. We also noted this in the caption: “Dots and bolded segments of the curves denote estimated mean ± SD parrotlet claw drag distances, respectively, for each surface if the foot did not slip (see Materials and methods).” The “E” refers to the expected value, which we also now clarify in the caption: “The expected value of the maximum tangential to normal force ratio is denoted as E[*F*_T,max_/*F*_N_].”

Figure 5 What does "highly non-linearly" mean? How is this evaluated? Need an objective measure for this characterization. Suggest simply "non-linearly".

Complied; we removed the word “highly.”

Unclear how 5B data are established.

Clarified; We clarified in the caption for Figure 5B that the data were measured in surface deformation tests and included a reference to the Materials and methods, which outline how the surface deformation tests were performed: “The depth of surface deformation due to claw penetration is small compared to claw geometry (see “Surface deformation tests” in Materials and methods).”

θmin is not clearly defined or illustrated in 5D – reference should be after 'angle', not after 'friction'; and "As shown for the Sweet Olive surface, smaller θmin amplify the effect of surface penetration…" is unclear to me how this amplifies surface penetration effect.

Complied; we changed the reference for θ_min_ to be after “angle.” Furthermore, we more clearly illustrated θ_min_ in Figure 5D and defined it in the main text as follows: “θ_min_ is the angle formed between the vertical axis and the net force vector on the claw (*F*_N_ + *F*_T_) when the tangential to normal force ratio is at its maximum.” Finally, we clarified how θ_min_ amplifies the surface penetration effect in the Figure 5 caption as follows: “Smaller θ_min_ amplify the effect of surface penetration, which increases usable surface, as shown by the growing discrepancy between the curves that account for surface penetration versus those that do not (DR = 0) for the Sweet Olive surface.”

"Specifically, θmin is the angle formed between a given normal force (FN) and the corresponding maximum tangential force vector (FT)." How can the angle between FN and FT not be 90 deg?

Clarified; Thank you for pointing this out. We replaced this statement with the clarification described in the previous response, “θ_min_ is the angle formed between the vertical axis and the net force vector on the claw (*F*_N_ + *F*_T_) when the tangential to normal force ratio is at its maximum.”

Figure 5F: silhouettes aren't mentioned in legend, or clear why they are shown.

Clarified; To address this, we now explain in the caption for 5F that the animal silhouettes show the approximate tip radii scales of those animals: “Thus when steeper surface features are required, smaller animals and perching robots with lower claw tip radii tend to have more usable surface than larger animals (silhouettes show approximate tip radii scale for associated animals and a perching aerial robot), but the trend reverses when shallow asperities are sufficient.”

Figure 6B and C are unclear to this reviewer. 6D and E are the most clear and relevant to this reviewer.

Clarified; To clarify Figure 6B and 6C, we added additional labeling that more clearly indicates the stable and slipping regions of the wrench space, as well as the effects of increasing torque.

Unclear why "structure strength (~length^2^)" Isn't this an area?

Clarified; To address this, we clarified that structure strength scales with length^2^ (squared) since stress scales with area: “However, isometric scaling also suggests that structure strength (~length^2^ since stress scales with area) diminishes with mass (~length^3^), so larger animals may be unable to achieve high safety margins during grasping.”

Also, it would help to reference methods here, re: how claw penetration depths were measured on different surfaces.

Complied; we added a reference to the Materials and methods for details on the claw penetration depth testing: “In particular, the traced surface model (Figure 5D-F) indicates that small animals like insects (beetle tip radius = 8-10 μm [Dai, Gorb and Schwarz, 2002]) or even robots, which generally rely on sharp fish hooks for “claws" (tip radius = 10-15 μm [Desbiens and Cutkosky, 2010]), have more usable surface on vertical surfaces (high F_T_/F_N_) (see Materials and methods for details on the claw penetration depth testing).”

Subsection “Friction drag tests”: This is also unclear: For friction drag tests, were claws and toe pad pulled along beam length or in circumferential direction? If the former this would presumably have different asperities/roughness measures. If the latter, explain more clearly and how/why surface curvature does not affect the analysis (were curvature effects removed beforehand?)

Complied; we clarified and explained this in the text as follows: “The use of the motor and linear slide enabled the claw and toe pad to be dragged consistently in the circumferential direction for all tests. […] Claws are very sensitive to hooking onto individual asperities (as compared to toe pads which have a larger spatial distribution of surface contacts), so to avoid missing asperities as a result of the system’s mass/momentum, the motor pulled the claw quasi-statically.*”*